# A low-threshold intervention to increase physical activity and reduce physical inactivity in a group of healthy elderly people in Germany: Results of the randomized controlled MOVING study

**Fabian Kleinke**[1,2]*, **Sabina Ulbricht**[2,3], **Marcus Dörr**[2,4], **Peter Penndorf**[1,2], **Wolfgang Hoffmann**[1,2], **Neeltje van den Berg**[1,2]

**1** Institute for Community Medicine, Section Epidemiology of Health Care and Community Health, University Medicine Greifswald, Greifswald, Germany, **2** DZHK (German Centre for Cardiovascular Research), Partner Site Greifswald, Greifswald, Germany, **3** Institute for Community Medicine, Section Prevention Research and Social Medicine, University Medicine Greifswald, Greifswald, Germany, **4** Department of Internal Medicine B, University Medicine Greifswald, Greifswald, Germany

* fabian.kleinke@uni-greifswald.de

## Abstract

### Background

Lack of physical activity (PA) and a high level of physical inactivity (PI) are associated with a higher risk for mortality and responsible for several non-communicable diseases including cardiovascular disease. Higher age is associated with a decrease of PA and an increasing level of PI. Studies have shown that interventions in the elderly have the potential to increase the amount of PA and to decrease the level of PI. However, most interventions are complex, elaborated, time- and resource-consuming. Here, we examined the effect of individual feedback-letters reporting the measured PA and PI in a sample of elderly people in Germany. Primary outcomes of the study were overall PA and PI after 6 months in the intervention group compared to a control group.

### Methods

We examined data from the MOVING intervention study (RCT) for people aged $\geq$ 65 years living in the northeast of Germany. At baseline, 3 and 6-months follow-up, all study participants wore a 3-axis accelerometer over a period of seven consecutive days. After the baseline measurement, the participants were randomized into intervention and control group. Participants in the intervention group received automatically generated, individualized feedback letters reporting their PA and PI by mail after the baseline measurement and after the 3-months follow-up. A Two-Way Mixed ANOVA with repeated measures was calculated with light, moderate and overall PA as well as PI as dependent variables, and group (between subject) and time (inner subject) as factors. The analysis based on retrospective data from the MOVING study (2016–2018).

**Data Availability Statement:** All relevant data are within the manuscript and its Supporting Information files.

**Funding:** The study was funded by the Federal Ministry of Education and Research (BMBF) as a site project of the German Centre for Cardiovascular Research (DZHK) (funding sign: 81Z7400174). The funders have had no influence on the conceptualization and conduct of the study and will not have any role in the data analysis and publication of the results.

**Competing interests:** The authors have declared that no competing interests exist.

## Results

N = 258 patients were recruited. N = 166 participants could be included in the analysis, thereof N = 97 women (58.4%). The mean age was 70.8 years (SD 4.8). At baseline, the participants had a mean wearing time of 5,934.5 minutes (SD = 789.5) per week, which corresponds to about 14 hours daily on average. The overall PA in the intervention group at the 6-months follow up was 2488.8 (95% CI 2358.9–2618.2) minutes and 2408.2 (95% CI 2263.0–2553.4) minutes in the control group. There was no statistically significant interaction effect (time*group) between the intervention and control group for the depending variables. Sensitivity analyses showed significant small positive effects of the interaction time*partnership, $F(2, 300) = 3.020$, $p = 0.05$, partial $\eta^2 = 0.020$.

## Discussion

On average, study participants had high levels of PA at baseline and showed a good adherence in wearing the accelerometer. Both is likely due to selection in the convenience study sample. Thus, some ceiling effect reduced the overall intervention effect somewhat. At baseline, the weekly average of PI was 3436.7 minutes, which correspondents to about 8.2 hours per day and about 57% of participants' daily waking time. The average level of PI could be slightly decreased in both study groups.

## Trial registration number

DRKS00010410, 17 May 2017.

## Introduction

There is strong evidence, that physical activity (PA) promotes healthy aging and that both lack of PA and a high level of physical inactivity (PI) are crucial risk factors for global mortality and several non-communicable diseases (NCD) [1–7].

PI is defined as "any waking behavior characterized by an energy expenditure ≤1.5 METs (metabolic equivalent of task) while in a sitting or reclining posture" [8, 9]. 31.1% of the world-wide adult population has a sedentary lifestyle, is insufficiently physically active and does not meet the recommendations for PA from the World Health Organization (WHO) [10]. Physical inactivity causes 3.2 million deaths per year worldwide. In 2010 PI was responsible for 69.3 million DALYs (disability-adjusted life years) [11]. PI causes a significant economic burden, in 2013, PI caused 53.8 billion US-Dollar healthcare costs worldwide [12]. As a consequence, the reduction of insufficient PA is a global target of the WHO [11, 13].

Depending on the definition, the proportion of PI across European countries ranges from 43.3% in Sweden up to 87.8% in Portugal. In Germany, 70.2% of men and 71.8% of women show a sedentary lifestyle [14]. High levels of moderate to vigorous physical activity (MVPA) are associated with a reduced risk for all-cause mortality [6, 15, 16]. In general, epidemiological evidence about positive effects of regular PA is strong and has accumulated over several decades [4, 17–19].

Several studies have shown that a physically active lifestyle (including promoting PA and minimizing PI) is associated with improved overall health status [4, 20] and a reduced risk for overall mortality by 22–34%, and CVD mortality by 27–35% [1]. In addition, regular PA is a

well-established method to prevent diabetes, hypertension and several types of cancer (i.e. breast and colon cancer) also for older people. Furthermore, regular PA reduces the risk of developing stroke about 25–30% [21]. Even people who are only 1–2 times per week physically active, have a reduced risk for mortality [22] compared to those with less PA.

There is strong evidence, that moderate-to-vigorous PA (MVPA) has positive effects on important health outcomes and is associated with a reduced risk of all-cause mortality [16]. Several international physical activity guidelines from the United States [23], United Kingdom [24], Australia [25], and from the WHO [26] promote the benefits of both moderate and vigorous PA. Recommendations for light PA are lacking in the guidelines [24–27]. However, increasing light PA also seems to be positively associated with relevant health outcomes, i.e. obesity and glucose metabolism, and is associated with a reduced risk for all-cause mortality [19, 28, 29]. In addition, light PA has beneficial effects on the health situation of elderly people, e.g. a higher cognitive performance [30], and it is positively associated with physical health and life satisfaction [31]. Hence, the impact of light PI on public health may be underestimated to some extent.

Increasing age is associated with decreasing regular PA, a decreasing number of daily steps [32] and an increase of PI which contributes to higher risks for chronic diseases (e.g. stroke) and multimorbidity [10, 33–36]. In the context of demographic change, particularly older people should be an important target group for extensive and practicable prevention strategies.

Interventions in the elderly have the potential to increase PA [37–39] and to reduce PI [40–42]. Several pedometer-based intervention studies have shown positive effects on the level of PA. Harris et al. showed a positive effect of an intervention consisting of four consultations by a primary care nurse on daily steps and weekly MVPA in ≥10 min bouts.

A meta-analysis on pedometer-based PA interventions pointed out that usage of a pedometer has a moderate positive effect on PA levels (average increase of 2,000 daily steps) [38]. These results correspondent to a systematic literature review in which participants in the intervention group increased their PA by 2,491 daily steps by using a pedometer-based intervention compared to the control group [39].

Interventions also have the potential to decrease PI [43]. Mutrie et al. have shown that a nurse-delivered intervention in the elderly (N = 41), based on social cognitive theory, reduced daily PI by 68 minutes in the intervention group (measured by pedometer and accelerometer) compared to the controls [44]. Another study of older adults (N = 478) showed that a home-based intervention based on social cognitive theory led to a decrease of 57 minutes of sitting time per day in the intervention group compared to the control group [42].

Most of the successful studies are based on complex interventions (i.e. personal consultations) [37]. These are associated with high costs and a large demand of personal resources. Dissemination of such interventions to the general older population may be hardly feasible, especially against the background of the demographic change. In conclusion, a translation of such interventions into daily life of a major proportion of the elderly has been difficult. Therefore, we designed a low-threshold and practicable intervention to increase overall PA and decrease PI.

## Research objectives

Aim of the study was the evaluation of a low-threshold intervention to improve overall PA and simultaneously reduce PI. The objective of this analysis was to examine the effect of a low-threshold intervention, consisting of automatically generated, individualized feedback-letters reporting the measured physical activity on PA and PI in a sample of apparently healthy elderly people in the Northeast of Germany (MOVING-study (Motivation-Oriented intervention

study for the elderly in Greifswald). Primary outcomes of the analysis were overall PA and PI after 6 months in the intervention group compared to the control group.

## Materials and methods

### Study sample

A detailed study protocol [45], as well as levels and determinants for PA and PI of the study sample of the MOVING study at baseline are published elsewhere [46]. In brief, the study population consists of people sampled from the general population who meet the following inclusion criteria:

- Age ≥ 65 years

- The possibility of being physically active in daily life

  Exclusion criteria were:

- Permanent use of a wheelchair (no ability to walk independently)

- Simultaneous participation in other studies addressing PA and PI

- Not accessible by telephone or cell phone

- Fulfillment of the WHO recommendations for PA (self-report) for people aged ≥ 65 at baseline (≥150 minutes of moderate PA or 75 minutes of vigorous PA)

All eligible patients were informed in detail about the study and had to give their written informed consent prior to study inclusion. The project data was assessed using eCRFs in an IT-supported documentation system [47]. At Baseline, all participants received a study ID-card with individual 1D-barcode for reliable re-identification.

### Study procedure

Potential participants and interested persons contacted the study staff by telephone. A short telephone interview was performed to check the inclusion and exclusion criteria. All persons who fulfilled the inclusion criteria were invited to the examination center of the German Centre for Cardiovascular Research (DZHK) of the University Medicine Greifswald. Here, all participants received a 3-axis accelerometer (ActiGraph GT3x-BT, Pensacola, FL, USA) and specific instructions on how to wear the accelerometer device.

After the baseline examination, participants were individually randomized 1:1 into the control and the intervention group. Follow-up examinations were conducted in both groups at 3 and 6 months after the baseline examination with the same program as in the baseline examination.

### Measures

All study participants received a physical examination consisting of blood pressure measurement and measurement of body weight, waist and hip circumference. Trained and certified study staff carried out all measurements according to standard operating procedures (SOP). At baseline, participants were asked for sociodemographic data (sex, age, education (number of years participants went to school)). In addition, participants received standardized paper-pencil questionnaires on PA and PI (IPAQ [48], German PAQ 50+ [49], MOST [50], SF8 [51] and SSA-scale [52]) that had to be completed after the 7-days of wearing the accelerometer device.

After the baseline and at the two follow-up examinations, all study participants wore the accelerometer device at daytime on the right hip. The participants were instructed to remove the device only for water-based activities. The three-axis accelerometer recorded PA, PI and sedentary breaks (a period of non-sedentary activity (e.g. walking) in between two sedentary conditions (e.g. sitting) continuously at a sampling frequency of 30 Hz over a period of seven consecutive days, starting at midnight after the examination day.

The software ActiLife 6.13.2–6.13.3 (ActiGraph Cop. ©, Pensacola, FL, USA) was used for data download and data processing. Raw data were calculated into 60-second epochs and saved as GT3X files.

We used specific cut points based on Freedson [53] to categorize the intensity of participants' PA. For adults, this cut-point model is considered as established for the determination of different physical activity categories [54]. Intensity of PA is categorized as sedentary (0–99 counts), light (100–1951 counts), moderate (1952–5724 counts), and vigorous (5725–9498 counts) PA. Non-wear time was defined as $\geq$ 60 min of continual zero counts.

## Intervention

The intervention based on individual accelerometer data and consisted of automatically generated, personalized feedback letters sent by mail. The participants in the intervention group received the letters after the baseline examination and the 3-month follow-up shortly after wearing the accelerometer. The participants in the control group received no feedback during the study period. According to the Declaration of Helsinki, we provided participants in the control group a detailed feedback about PA and PI after the end of the study within a thank you letter. This procedure ensured that all participants (both groups) received in sum the same information at the end of the study.

The feedback letters contained three graphs, reporting on the measured PA and PI, with provision of the number of steps per day, daily and weekly moderate and vigorous PA in minutes and daily and weekly sedentary time in minutes, as well as a paragraph about the importance of sedentary breaks. The participants received age appropriate recommendations for PA based on the WHO-guidelines [26]. The feedback letters were automatically generated in R software (version 3.3.2, Lucent Technologies, Murray Hill, NJ, USA).

## Data analysis

To examine the effects of the intervention, we analyzed overall, light and moderate PA as well as PI as dependent variables with a Two-Way Mixed ANOVA with repeated measures with group (between subject) and time (inner subject) as factors. For baseline data, we analyzed the equality of variables between both study groups. Due to significant difference between both groups in partnership (p = 0.006), we included partnership as a covariable in the Two-Way Mixed ANOVA. First, we analyzed the assumptions and requirements of mixed ANOVA i.e. distribution and homoscedasticity and detected no violation of the assumptions". Bonferroni adjustment was made because several analyses were performed.

All statistical analyses were performed using pseudonymized data. In addition, we performed a sub-group analysis and excluded participants with $\geq$ 150 minutes of weekly moderate PA or $\geq$ 75 minutes of weekly vigorous PA at baseline (according to the WHO recommendation).

A valid measurement day was defined as a record of at least 10 hours of total wearing time during the day. Additionally, a record time of at least 4 valid days was required for all data analyses.

Statistical significance was assumed for p-values <0.05. Data processing and statistical calculations were performed with IBM SPSS Statistics 26 or later (1989–2020 by IBM Corp. ©, Armonk, New York, USA.).

### Ethical consideration

In 2016, the clinical ethics committee of the University Medicine Greifswald (protocol number BB071/16) approved the study.

## Results

A number of 258 potential participants were contacted personally or by telephone, thereof 206 were included in the study and randomized (Fig 1).

At baseline, the participants had a mean wearing time of the accelerometer of 5,934.5 minutes (SD = 789.5) per week, which corresponds to 14.1 hours daily. N = 166 participants could be included in the analysis, thereof n = 97 women (58.4%) and n = 69 men (41.6%). At baseline, the mean age was 70.8 years (SD = 4.8) (Table 1).

At baseline, the average time of overall PA was 2473.7 (95% CI 2387.0–2560.4) minutes per week, to which light PA contributed the largest part with 2260.9 (95% CI 2181.1–2340.7) minutes. In addition, the average of weekly PI (baseline) was 3460.8 (95%-CI 3337.5–3584.0) minutes per week which correspondents to about 8.2 hours per day and about 57% of participants' waking time.

Participants in the intervention group slightly increased their weekly time of light PA from baseline to 6-months follow-up on average by + 54.0 minutes (control group—59.2 minutes). The average overall PA increased in the intervention group by 25.0 minutes per week (control group—75.9 minutes) between baseline and 6 months follow-up. At 6-months follow-up (primary endpoint), the weekly difference of overall PA between intervention group (2488.8 minutes, 95% CI 2358.9–2618.2) and control group (2408.2 minutes, 95% CI 2263.0–2553.4) was 80.6 minutes which correspondents to about 1.3 hours of additional overall PA per week in the intervention group (Table 2).

At baseline, both study groups started almost on the same value of overall PA. At 6-months follow-up, the amount of overall PA decreased in the control group (red line) the baseline value while participants in the intervention group (blue line) could slightly increase their weekly overall PA (Fig 2).

At baseline, n = 97 (58.4%) of the study participants (both study groups) achieved the international recommendations for moderate PA ($\geq$150 minutes moderate or $\geq$ 75 minutes vigorous PA). The proportion of participants who fulfilled the recommendations slightly decreased over study time (Table 3).

There was no statistically significant interaction between time and group (interaction effect: time*group) for the depending variables PI, light, moderate and overall PA (sphericity assumed). P-values were ranging between 0.317 to 0.546. *Physical inactivity (n = 146)*

The results of the Two-Way Mixed ANOVA showed that there was a significant main effect of group, $F(1, 143) = 5.335$, $p = 0.022$, partial $\eta^2 = .036$. In addition, there was no significant main effect of time, sphericity assumed $F(2, 286) = 0.763$, $p = 0.467$, partial $\eta^2 = .005$.

Descriptive statistics showed that both study groups decreased the amount of PI over study time. At 6-months follow-up, participants included in the analyses had an amount of 3310.4 minutes (SD = 536.1) of PI (mean) in the intervention group, and 3161.1 minutes (SD = 581.8) of PI (mean) in the controls. Besides that, the differences between both groups were about the same amount of minutes in PI at each measurement point (baseline, 3- and 6-months follow-up).

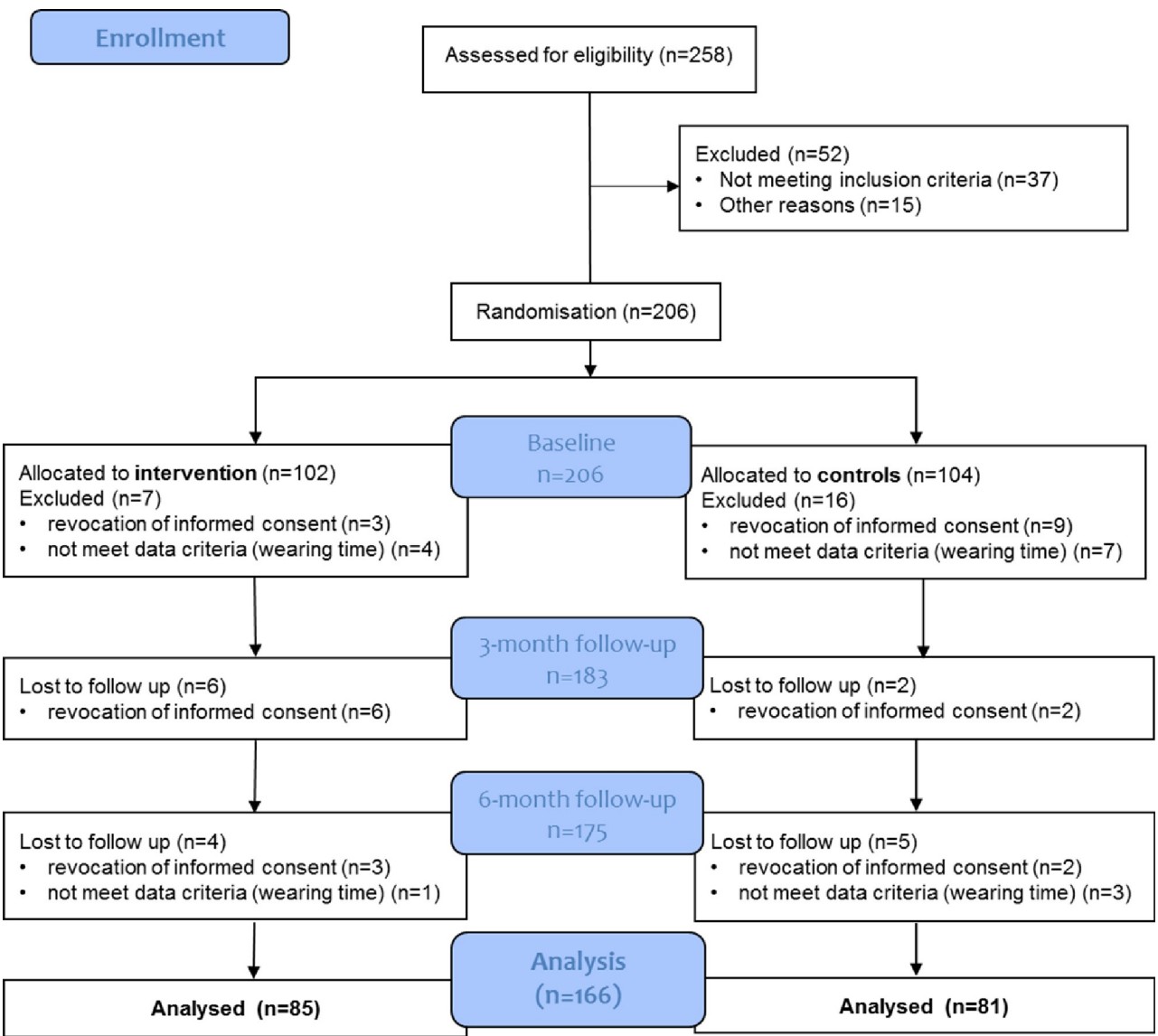

**Fig 1. Consort diagram: Recruited participants and participants included in the analysis (N = 166).**

### Light PA (n = 155)

The results of the Two-Way Mixed ANOVA showed that there was no significant main effect of group, $F(1, 153) = 0.017$, $p = 0.895$, partial $\eta^2 = .000$, and time, sphericity assumed $F(2, 304) = 2.196$, $p = 0.113$, partial $\eta^2 = .014$.

Descriptive statistics showed almost the same baseline value for light PA between the intervention (2235.8 minutes (SD = 495.7) and control group (2232.1 minutes, SD = 536.4). After an increase of light PA in both study groups to 3-months follow-up, participants in the control group decreased minutes of light PA even lower than the baseline value (2187.4 minutes, SD = 604.5) at 6-months follow-up. Besides that, participants in the intervention group remained almost at the same amount of light PA at 6-months follow-up (2298.8 minutes, SD = 557.8). Thus, participants in the intervention group showed a higher amount of light PA

**Table 1. Descriptive characteristics of the study sample (N = 166), complete and valid data sets (baseline, 3 and 6 months follow-up).**

| Characteristics | n | Control group Mean (SD) or n (%) | n | Intervention group Mean (SD) or n (%) | n | Both groups Mean (SD) or n (%) |
|---|---|---|---|---|---|---|
| Sex (women) | 81 | 47 (58.0%) | 85 | 50 (58.8%) | 166 | 97 (58.4%) |
| Age (yr) | 81 | 71.2 (SD 5.0) | 85 | 70.4 (SD 4.6) | 166 | 70.8 (SD 4.8) |
| Number of participants currently living in a partnership (yes)* | 77 | 47 (61.0%) | 83 | 67 (80.7%) | 160 | 114 (71.3%) |
| Education (yr) | 74 | | 82 | | 156 | |
| < 10 years | | 14 (18.9%) | | 18 (22.0%) | | 32 (20.5%) |
| = 10 years | | 25 (33.8%) | | 17 (20.7%) | | 42 (26.9%) |
| > 10 years | | 30 (40.5%) | | 45 (54.9%) | | 75 (48.1%) |
| Other | | 5 (6.8%) | | 2 (2.4%) | | 7 (4.5%) |
| Body mass index (kg/m²) | 81 | | 85 | | 166 | |
| < 25 | | 22 (27.2%) | | 23 (27.1%) | | 45 (27.1%) |
| ≥ 25 and < 30 | | 33 (40.7%) | | 30 (35.3%) | | 63 (38.0%) |
| ≥ 30 | | 26 (32.1%) | | 32 (37.6%) | | 58 (34.9%) |
| Waist circumference (cm) | 81 | 96.7 (SD 13.1) | 85 | 94.8 (SD 14.6) | 166 | 95.8 (SD 13.9) |
| Hip circumference (cm) | 81 | 104.1 (SD 11.8) | 85 | 102.7 (SD 10.4) | 166 | 103.4 (SD 11.1) |
| Blood pressure (sys/dia mmHg) (Pulse /min) | 81 | 133/74 (70) | 84 | 132/72 (73) | 165 | 133/73 (71) |
| Wearing time of the accelerometer | | | | | | |
| Baseline | 81 | 5,922.7 (SD 926.8) | 85 | 5,945.7 (SD 637.2) | 166 | 5,934.5 (SD 789.5) |
| 3-month follow-up | 81 | 5,833.6 (SD 839.1) | 84 | 5,826.4 (SD 683.9) | 165 | 5,829.9 (SD 761.7) |
| 6-month follow-up | 79 | 5,636.4 (SD 719.7) | 82 | 5,840.7 (SD 719.0) | 161 | 5,740.5 (SD 724.4) |

Notes: *n* number of subjects, *SD* standard deviation *significant difference between intervention and control group (p = 0.006).

at 6-months follow-up (primary endpoint) compared to the controls (+111.4 minutes of light PA) in the descriptive statistics.

## Moderate PA (n = 147)

The results of the Two-Way Mixed ANOVA showed that there was no significant main effect of group, $F(1, 144) = 1.426$, $p = 0.234$, partial $\eta^2 = .010$, and time, sphericity assumed $F(2, 288) = 0.715$, $p = 0.490$, partial $\eta^2 = .005$.

Descriptive statistics showed a constant decline of moderate PA in the control intervention group starting from baseline (172.4 minutes, SD = 124.1) to 6-month follow-up (145.9 minutes, SD = 115.1). Participants in the intervention group started with an average amount of moderate PA about 201.8 minutes (SD = 149.6) and could increase the amount after a decline at 3-months follow-up (163.9 minutes, SD = 127.4) at 6-months follow-up to 170.9 minutes (SD = 126.3).

## Overall PA (n = 155)

The results of the Two-Way Mixed ANOVA showed that there was no significant main effect of group, $F(1, 152) = 0.005$, $p = 0.945$, partial $\eta^2 = .000$, and time, sphericity assumed $F(2, 304) = 1.638$, $p = 0.196$, partial $\eta^2 = .011$.

Descriptive statistics showed slightly increase (+36.1 minutes) of the amount of overall PA in the intervention group starting at 2459.6 minutes (SD = 558) to 3-months follow-up (2495.7, SD = 564.4). Participants in the intervention group stayed almost on this level at 6-months follow-up (2494.6, SD = 590.8). The controls started almost on the same baseline value (2459.0, SD = 581.1). After an increase to 3-month follow-up (2504.7, SD = 657.2), the

**Table 2. Physical activity and inactivity (mean number of minutes during the wearing-week), by intensity of physical activity and overall PA (N = 166).**

| Study phase | Study Group | Physical inactivity | Sedentary breaks | Light PA | Moderate PA | Vigorous PA | Overall PA | Steps Counts |
|---|---|---|---|---|---|---|---|---|
| | | Mean (CI 95%) | Mean (CI 95%) | Mean (CI 95%) | Mean (CI 95%) | Mean (CI 95%) | Mean (CI 95%) | Mean (CI 95%) |
| **Baseline** | Intervention | 3482.0 | 97.6 | 2246.4 | 213.6 | 3.8 | 2463.8 | 99,278.0 |
| | group (n = 85) | (3336.1–3627.8) | (92.7–102.4) | (2139.9–2352.8) | (180.6–246.6) | (0.6–7.0) | (2342.8–2584.8) | (92,889.8–105,666.2) |
| | Control | 3438.5 | 95.9 | 2276.1 | 201.1 | 6.6 | 2484.1 | 97,763.2 |
| | group (n = 81) | (3234.3–3642.7) | (89.3–101.9) | (2154.4–2397.8) | (166.5–235.6) | (1.2–14.3) | (2357.2–2611.1) | (90,840.4–104,685.9) |
| | *Total (n = 166)* | 3460.8 | 96.8 | 2260.9 | 207.5 | 5.1 | 2473.7 | 98,538.8 |
| | | (3337.5–3584.0) | (93.0–100.6) | (2181.1–2340.7) | (183.8–231.1) | (1.1–9.2) | (2387.0–2560.4) | (93,884.7–103,193.0) |
| **3-months follow-up** | Intervention | 3351.1 | 92.4 | 2292.3 | 180.3 | 2.7 | 2475.3 | 99,554.3 |
| | group (n = 84) | (3204.8–3497.3) | (87.6–97.0) | (2174.7–2409.8) | (148.3–212.3) | (0.2–5.6) | (2348.7–2602.0) | (93,268.3–105,840.2) |
| | Control | 3318.8 | 93.6 | 2327.5 | 181.5 | 5.3 | 2514.8 | 97,567.4 |
| | group (n = 81) | (3140.0–3497.7) | (87.8–99.4) | (2188.9–2466.2) | (151.0–212.0) | (1.0–9.7) | (2369.9–2659.6) | (90,803.8–104,330.9) |
| | *Total (n = 165)* | 3335.3 | 93.0 | 2309.6 | 180.9 | 4.0 | 2494.7 | 98,578.9 |
| | | (3221.3–3449.2) | (89.4–96.6) | (2219.9- | (159.0–202.8) | (1.4–6.6) | (2399.7–2589.7) | (94,013.7–103,144.1) |
| **6-months follow-up** | Intervention | 3354.8 | 92.5 | 2300.4 | 184.9 | 3.3 | 2488.8 | 99,613.6 |
| | group (n = 83) | (3210.0–3499.6) | (88.1–96.9) | (2180.1–2420.8) | (154.2–215.6) | (0.7–5.8) | (2358.9–2618.2) | (93,526.0–105,701.2) |
| | Control | 3228.2 | 90.2 | 2216.9 | 187.7 | 3.5 | 2408.2 | 94,301.4 |
| | group (n = 79) | (3087.0–3369.4) | (85.6–94.7) | (2077.8–2356.0) | (150.3–225.1) | (1.6–8.5) | (2263.0–2553.4) | (87,290.4–101,312.5) |
| | *Total (n = 162)* | 3293.1 | 91.4 | 2259.7 | 186.3 | 3.4 | 2449.5 | 97,023.1 |
| | | (3192.5–3393.7) | (88.2–94.5) | (2168.8–2350.6) | (162.5–210.1) | (0.6–6.2) | (2353.2–2545.8) | (92,426.6–101,619.7) |

Notes: *n* number of subjects, *CI* 95% confidence interval.

amount of overall PA declined to 6-months follow-up below the intervention group (2399.4, SD = 661.4). At 6-months follow-up, the group difference was 95.2 minutes of weekly overall PA.

## Sensitivity analyses

In a sensitivity analysis, we excluded participants with ≥ 150 minutes of weekly moderate PA or with ≥ 75 minutes of vigorous PA at baseline (according to the WHO recommendation). Here (n = 153), we found a statistically significant interaction effect for time*partner (covariable) for light PA, sphericity assumed $F_{(2, 300)} = 3.020$, $p = 0.05$, partial $\eta^2 = 0.020$. This result indicate a small positive effect on light PA over study time in participants who are in a partnership.

## Discussion

The results of this analysis show that the levels of light, moderate and overall PA in our sample were exceptionally high for this age group [55–57]. Both groups started on almost the same values of overall PA at baseline (intervention group = 2463.8 (95% CI 2342.8–2584.8) minutes,

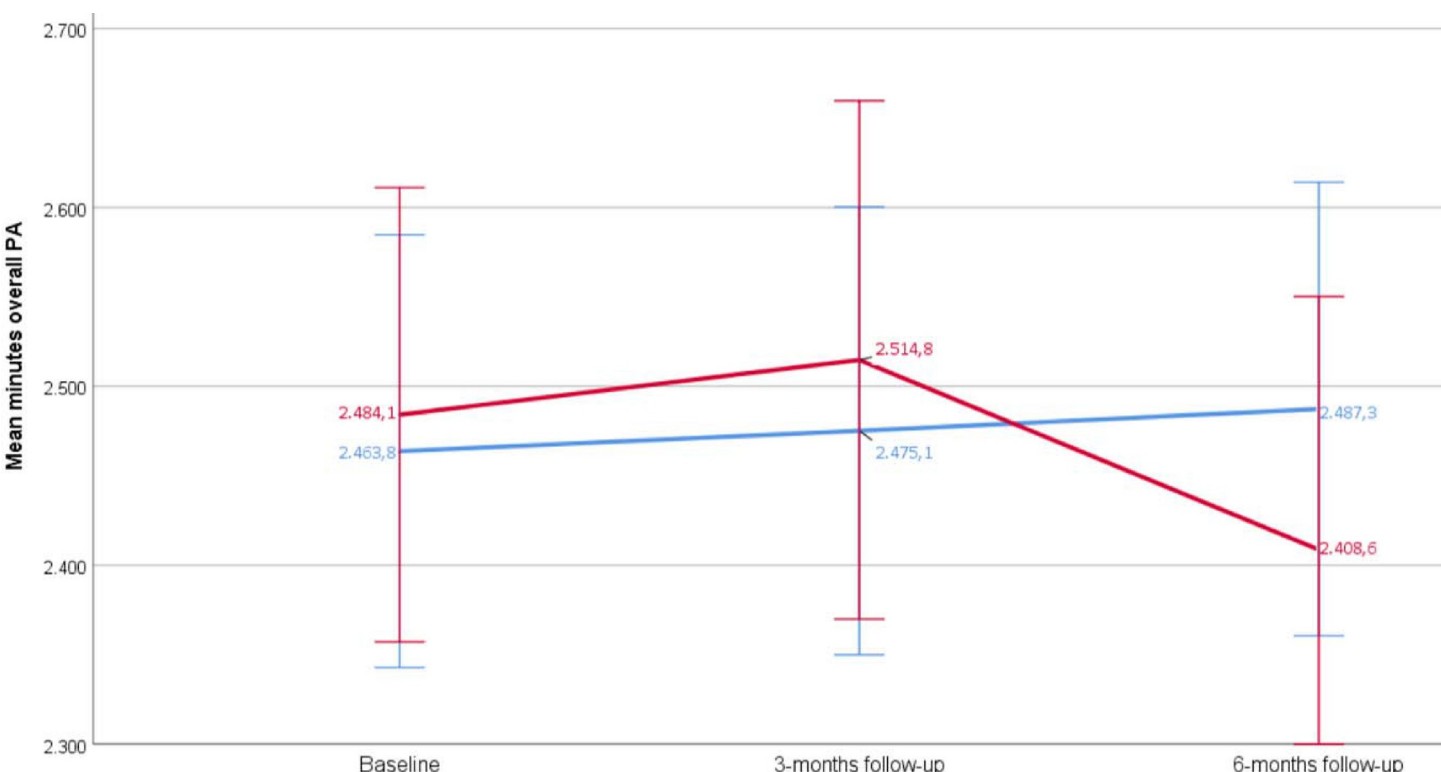

**Fig 2.** Mean minutes of overall PA (95% CI, intervention group and control group) at baseline, 3 and 6 months follow-up (N = 166), control group (red line), intervention group (blue line).

control group = 2484.1 (95% CI 2357.2–2611.1) minutes. The level of overall PA over both groups remained almost on the same level at the 3-months follow-up (intervention group = 2475.3 (95% CI 2348.7–2602.0) minutes, control group = 2514.8 (95% CI 2369.9–2659.6) minutes), and 6-months follow-up (intervention group = 2488.8 (95% CI 2358.9–2618.2) minutes, control group = 2408.2 (95% CI2263.0–2553.4) minutes). At 6-months follow-up, participants in the intervention group showed more overall PA compared to the participants in the control group (difference = 80.6 minutes per week which correspondents to about 1.3 hours of additional overall PA per week in the intervention group).

In previous randomized controlled intervention trials promoting PA increasing PA levels have often been observed also in the control groups [58, 59]. Therefore, several possible

**Table 3. Number and proportion of study participants who fulfilled the WHO recommendations PA for people aged over 65 years, separately for intervention and control group (N = 166).**

| WHO recommendation | Study group | Baseline (n = 166, IG = 85, CG = 81) | 3-month follow-up (n = 165, IG = 84, CG = 81) | 6-month follow-up (n = 162, IG = 83, CG = 79) |
|---|---|---|---|---|
| **moderate PA ≥150 min. per week or** | Intervention group | 49 (57.6%) | 43 (51.2%) | 41 (49.4%) |
| **vigorous PA ≥75 min. per week** | Control group | 48 (59.3%) | 39 (48.1%) | 38 (48.1%) |
| | Total | 97 (58.4%) | 82 (49.7%) | 79 (48.8%) |

n number of subjects, fulfillment of WHO recommendations for moderate PA for people aged 65 and older: ≥150 minutes moderate PA or ≥75 minutes vigorous PA or an equivalent combination per week (2 minutes moderate PA correspond to 1 minute vigorous PA).

reasons have been proposed, e.g. the "Hawthorne effect", which may have changed participants' behavior as a result of participating in the study–irrespective of their randomisation. In our study, we observed this effect according to the amount of PI, both study groups (including controls) decreased PI over study time.

However, in a longitudinal analysis at the 6-month follow-up, the levels of light and overall PA had decreased in the control group (below the baseline value), while PA in the intervention group remained almost at the same level. Seasonal influences, (weather) cannot be the cause for the difference, because the intervention and the control group were simultaneously recruited and included between November 2016 and December 2018.

Also the amount of PI was high and with 3460.8 (95% CI 3337.5–3584.0) minutes per week at baseline, which corresponds to 8.2 hours/day and is comparable with other studies examining the same age group [14, 60]. Thus, participants in both study groups spent on average about 57% of their waking time in PI at baseline. The high amount of PI combined with a parallel high amount of PA indicates that PI and PA can be seen as independent factors.

The intervention showed no significant effect regarding the primary outcomes. However, a large part of the participants was already physically active on a high level: 58.4% (n = 97) of the participants already met the recommendations for people aged over 65 years of the WHO at baseline. This limits the potential to increase the participant's PA. Sensitivity analyses excluding very active participants, showed significant effects of the intervention on overall PA among the subgroup of the less active. Thus, further research should focus more on people with low PA levels. However, as a consequence of some self-selection of the more physically active elderly, recruitment of people with a low level of PA may be challenging.

The layout of interventions to increase PA and reduce PI is crucial. As previous research has shown, interventions should focus on the target group and especially on individual factors (e.g. age, sex) [46]. In addition, an inclusion of environmental and interpersonal factors can be helpful to change participants' behavior [46].

The study participants showed a good adherence to wearing the accelerometer. At baseline, the mean wearing time was on average 5934.5 minutes (SD = 789.5), which correspondents to 14.1 hours per day. Only n = 15 participants had to be excluded from the analysis because of too little wearing time over the six months follow-up time. In conclusion, our data support that the use of an accelerometer is feasible and practical in this age group.

Previous research has shown that intervention studies have the potential to increase PA and decrease PI [37–42]. Most of the previous studies are characterized by rather complex and time-consuming interventions, e.g. personal consultations or telephone calls, which limits the transfer of successful concepts to the general population. Our intervention used individualized feedback-letters based on measured PA and PI and was developed to examine a low-threshold intervention with high practicability and a high potential for translation into the general population. Due to some ceiling effect, the overall intervention effect reduced somewhat.

Especially elderly people are a relevant target group for promoting light PA, because an increase of moderate and vigorous PA might be harder to achieve for this age group [61]. Presently, quantitative recommendations towards PI are lacking and the public health message comes down to "*move more at any intensity*" [6, 15]. Thus, light PA should be included in global recommendations [62].

Several studies have shown that effective interventions to increase PA consider behavioral-change aspects and address participants' needs and beliefs. Thus, targeting and tailoring are two important aspects of successful interventions [63].

The number of intervention studies promoting PA and reducing PI increased continuously over the last decade. While there is international consensus regarding PA recommendations, it is too early to implement quantified recommendations regarding PI, although some countries

(e.g. Australia) introduced general guidelines to reduce sitting time in general and break up long periods of sitting regularly [25]. In addition, there is an ongoing discussion, whether PI is an independent risk factor for mortality [64], and there is still no global unified definition for PI. Thus, further research regarding PI and specific and quantitative recommendation are urgently needed.

## Limitations and strengths

We recruited study participants in a convenience sample using a variety of recruitment methods. Some of these methods have likely increased the proportion of participants with above average PA compared to PA in the general population–so the composition of our study sample somewhat limits the generalizability of our results. Due to this fact, a selection bias can not be excluded. Prior to recruitment, we tried to exclude very active people from the study (based on self-assessment) to address a possible ceiling effect. As the results have shown, many participants underestimate their own amount of PA. Therefore, we calculated a sensitivity analysis excluding very active participants with an amount of ≥150 minutes of moderate PA. Some ceiling effect reduced the overall intervention effect somewhat.

In general, accelerometry is an internationally established method for the objective assessment of physical activity and allows a valid and reliable record of PA intensity, frequency, and duration [54, 65]. Although, currently no single specific device can be recommended for all purposes, since the choice of the device depends on the research question, design and target group [54]. However, data from the accelerometer device can potentially differ from the real levels of PA and PI especially in the elderly, because several activities are carried out in standing (i.e. gardening) or sedentary positions (i.e. gymnastic on stools) which cannot be assessed accurately. Only few missings were obtained for the dependent variables (n = 4).

## Conclusion

Global demographic changes are associated with a large increase of the number of older people and a simultaneous growth of non-communicable-chronic diseases (i.e. cardiovascular disease). Therefore, there is a high need for effective prevention strategies. Although we detected no significant differences in the primary outcome in the interaction effect (time*group) between the study groups, descriptive statistics give a hint of a small positive effect of the intervention. Sensitivity analysis has shown, that partnership is a significant positive factor for light PA (time*partnership). Thus, further research should focus on the recruitment of the adequate target groups (especially people without partner and low levels of PA and high levels of PI). Therefore, local stakeholders should be consulted and setting related factors should be considered.

Most of the successful studies are based on complex and/or resource consuming interventions (i.e. personal consultations), which is accompanied by high costs and a large amount of personal resources. In the future, the usage of internet-based communications and e-health applications (e.g. video-consultation) can support the provision of information and interventions, which makes a translation of (complex) interventions into daily life easier.

Results of our study confirm that a low-threshold intervention with low costs and little amount of personal resources can be implemented and is able to hold up adherence of elderly people. Our analyses shows limited and not significant, but seems to have a small positive effect of a low-threshold intervention after 6 months follow-up.

## Supporting information

**S1 Table. Descriptive characteristics of the study sample (N = 166), complete and valid data sets (baseline, 3 and 6 months follow-up).** Notes: *n* number of subjects, *SD* standard

deviation.
(PDF)

**S2 Table. Physical activity and inactivity (mean number of minutes during the wearing-week), by intensity of physical activity and overall PA (N = 166).** Notes: n number of subjects, CI 95% confidence interval, M mean.
(PDF)

**S3 Table. Number and proportion of study participants who fulfilled the WHO recommendations PA for people aged over 65 years, separately for intervention and control group (N = 166).**
(PDF)

**S1 File. Dataset.**
(XLSX)

## Author Contributions

**Conceptualization:** Fabian Kleinke, Sabina Ulbricht, Marcus Dörr, Wolfgang Hoffmann, Neeltje van den Berg.

**Formal analysis:** Fabian Kleinke.

**Project administration:** Fabian Kleinke.

**Software:** Peter Penndorf.

**Supervision:** Wolfgang Hoffmann, Neeltje van den Berg.

**Writing – original draft:** Fabian Kleinke.

**Writing – review & editing:** Sabina Ulbricht, Marcus Dörr, Wolfgang Hoffmann, Neeltje van den Berg.

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
