## [Decision Letter · Decision Letter 0]

17 Mar 2021

PONE-D-21-04633

A low-threshold intervention to increase physical activity and reduce physical inactivity in a group of healthy elderly people in Germany: Results of the randomized controlled MOVING study

PLOS ONE

Dear Dr. Kleinke,

Thank you for submitting your manuscript to PLOS ONE. After careful consideration, we feel that it has merit but does not fully meet PLOS ONE’s publication criteria as it currently stands. Therefore, we invite you to submit a revised version of the manuscript that addresses the points raised during the review process.

Even though some reviewers consider that major revision is required and others consider that the manuscript should be rejected, I have decided not to reject it due to the relevance of the problem under study.

In the Introduction, there is redundant information that could be condensed. Is it necessary to talk about PA and PI? My question is because if they are opposites, only one of these terms should be used.

The objective of the treatment (increasing PA) must be differentiated from the objective of the study (exploring the effectiveness of the treatment).

In the Methods section, it does not seem to be clear how the sample size is N = 157; even one reviewer comments that the initially defined inclusion criteria are being violated.

The authors say: "To examine the effects of the intervention, we analyzed the overall, light, moderate and vigorous PA and PI after 6 months in the intervention group compared to the control group". These qualitative levels of PA have not previously been defined.

The difference between intervention and non-intervention should be made explicit. Was it only that the experimental group received the letters and the control group did not receive them? According to the Declaration of Helsinki regarding placebo use, what was done with the control group at the end of the study?

Please specify what the variables "education" and "health status" measure.

I agree with the reviewers that statistical analysis is not appropriate. Reviewer 2 suggests applying ANOVAs and considers it of interest to explore the effect of other variables (age, sex, etc.). Please report differences in age, sex, years of education, BMI, waist and hip circumference, blood pressure, etc., between groups; maybe it will be necessary to include some of these variables as covariables to mitigate their effects.

I also agree with Reviewer 2 concerning some conclusions that are not supported by the results.

We look forward to receiving your revised manuscript.

Kind regards,

Thalia Fernandez, Ph.D.

Academic Editor

PLOS ONE

Journal Requirements:

Reviewers' comments:

Reviewer's Responses to Questions

**Comments to the Author**

1. Is the manuscript technically sound, and do the data support the conclusions?

Reviewer #1: Partly

Reviewer #2: No

Reviewer #3: Yes

2. Has the statistical analysis been performed appropriately and rigorously? 

Reviewer #1: Yes

Reviewer #2: No

Reviewer #3: No

3. Have the authors made all data underlying the findings in their manuscript fully available?

Reviewer #1: Yes

Reviewer #2: No

Reviewer #3: Yes

4. Is the manuscript presented in an intelligible fashion and written in standard English?

Reviewer #1: Yes

Reviewer #2: Yes

Reviewer #3: Yes

5. Review Comments to the Author

Reviewer #1: I will focus on methods and discussion

Major

1) As the authors acknowledge this is a select population that makes generalisability of the findings questionable.

2) Some information on the randomisation procedure and how it was implemented is needed.

3) The sample size has not been arrived to through a power calculation but it is a convenience sample.

4) Results are poorly reported. Specifically the authors needs to make clear:

a) Drop outs - from 206 to 157?

b) report characteristics by group (i.e. Table 1). was balance across these key parameters achieved?

c) if not (i.e. b) then a t-test is not the apprpriate analysis

e) a secondary outcome should have been time wearing the device and that should have been reported in the abstract as well

f) report drop outs by group. Also the number of drop outs is relatively large for a keen sample of healthy adults.

5) No mention of missing data. was multiple imputation considered to boost the sample and get more robust estimates, compared to a complete case analysis?

Minor

1) Abstract, discussion: no clear mention of the intervention and why it may not work.

2) A more deterministic approach could have been used to assign people to groups, something often used when samples are modest, as they are here (to ensure good balance on key characteristics).

Reviewer #2: Authors evaluated the effectiveness of a low-threshold intervention to increase PA in elderly. Two groups (intervention-control) were recruited. No differences in most of the variables were found. Authors concluded that PA could be improved in participants with low levels of PA in both groups of this low-threshold intervention.

Several concerns regarding the rationale, methodology, results and discussion are explained below.

Authors propose “targeting and tailoring are two important aspects of successful interventions” on basis of the participants’ needs and beliefs, however, what they propose here “a low-threshold intervention” does not meet this criterion. Please justify and show evidence (if the case) of the effectiveness of a more “practicable intervention”, maybe from studies were only biofeedback is used to increase PA.

Table 1. Why the number of participants (n) change across the descriptive characteristics, for instance, there are more (161) participants in the line “Number of participants

currently living in a partnership (yes)” than those included in the study.

Please, include a Table showing the “descriptive characteristics” between groups and the statistical comparisons to exclude the presence of significant differences in these variables.

Explain, in the exclusion criteria the cutoff for the fulfillment of the PA according to WHO that you considered. In this regard, if this was an exclusion criterion, why the whole sample was included in the first analysis, and just excluded for the subsequent subgroup analysis?

Explain, in data analysis, how delta values were calculated.

At methodological level, I see two main problems, 1) participants with ≥ 150 minutes of weekly moderate PA at baseline (according to the WHO recommendation) which actually fulfillment an exclusion criterion, were included. 2) The statistical approaching is not justified for this data, why to use t-test instead of two way anova with group (2) and follow-up (3) as factors? The latter could help you to understand within subjects and between groups differences. On the other hand, why data of the first follow-up is not analyzed but included in Tables and Figures?

The only significant result was a difference in delta value between groups, however, this result did not confirm a PA increase due the intervention, but a decrease in the PA in the control group, according with Figure 2, is that right? Therefore, the effectiveness of the intervention is not corroborated?

In the first paragraph of the discussion, authors stated that the intervention group showed more overall PA compared to the participants in the control group, however this result was not significant. Therefore, no differences between groups were observed at any time.

In lines 286-287 authors suggest “Interventions should focus on the target group and especially on individual factors (e.g. age, sex)”, but not analysis was carried out with these variables in this study, it could be an interesting approaching. It is known, in previous literature, the differences between sexes in terms of PA and self-efficacy to PA.

From lines 301-313 authors discuss the importance of light PA increase and PI decrease, but their results did not show evidence of the effectiveness of the intervention even for these levels of PA.

In the conclusion authors suggest “Low-threshold interventions have the potential to increase PA and reduce PI in the general elderly population” however their results did not support this statement.

Reviewer #3: The paper requires a new organization of the introduction section. It is hard to read. Additionally, the statistical analysis and the results reported seem to be insufficient for the study's aims. I suggest major revision based on these aspects.

6. PLOS authors have the option to publish the peer review history of their article (what does this mean?). If published, this will include your full peer review and any attached files.

Reviewer #1: No

Reviewer #2: No

Reviewer #3: **Yes: **Graciela Catalina Alatorre-Cruz

---

## [Author Response · Author response to Decision Letter 0]

29 Apr 2021

PONE-D-21-04633

A low-threshold intervention to increase physical activity and reduce physical inactivity in a group of healthy elderly people in Germany: Results of the randomized controlled MOVING study

PLOS ONE

Response to Reviewers

Thank you for coordinating the review process and for your constructive comments to the manuscript. In our opinion, the comments help to increase the quality of the manuscript, especially the part of the statistical analyses. We have revised the manuscript and addressed all reviewer’s comments. 

Comment: In the Introduction, there is redundant information that could be condensed. Is it necessary to talk about PA and PI? My question is because if they are opposites, only one of these terms should be used.

Response: Thank you for the comment. To our knowledge, “we still know little about the independent detrimental health effects of sitting, and the possibility that sitting is mostly the inverse of physical activity remains”. (Stamatakis, 2018a). In addition, the evidence about sedentary behavior is “underdeveloped and inconsistent; it cannot support quantitative guidance” yet. 

Literature points out, that PA is not “only” the opposite of PI. Rather, PA and PI are in an interplay that has not yet been finally clarified (Ekelund, 2018, Stamatakis, 2018b). In our opinion, it is therefore important to analyze both factors and to mention them in the introduction. In sum, “Understanding the synergistic and independent effects of (ideally objectively measured) MVPA, standing, light intensity PA and sitting on CVD risk is a future research priority” (Stamatakis, 2018b). For this reasons, it is necessary to talk about both, PA and PI. 

Comment: The objective of the treatment (increasing PA) must be differentiated from the objective of the study (exploring the effectiveness of the treatment).

Response: Thank you for your comment. In the revised version of the manuscript, we have addressed your comment and emphasized the difference between the treatment and the objective of the study more clearly at the corresponding points in the manuscript. The objective of the study is the evaluation of a low-threshold intervention to improve PA and reduce PI among older study participants. The objective of the treatment is to improve PA and reduce PI. (please see lines 135 - 136)

Comment: In the Methods section, it does not seem to be clear how the sample size is N = 157; even one reviewer comments that the initially defined inclusion criteria are being violated.

Response: Thank you for this comment. For the traceability of the dropouts for each study group, we provide a Consort diagram (Fig. 1) according to PLOS ONE requirements. The Consort diagram is an attached file (supporting information) – please see Fig. 1 “S1 Fig. Consort diagram: Recruited participants and participants included in the analysis (n = 166)”. 

After a carefully check of the data, we could increase the number of complete data sets about n = 5. Due to Intention-to-treat principle, we also included n = 4 incomplete datasets. In sum, we could increase the study sample about n = 9, so the study sample in the revised manuscript is now n = 166 at baseline. Due to the small number of incomplete data sets at Follow-up 1 (n=1) and Follow-up 2 (n=3), we decided us against an imputation of data. For the main analysis (Two-Way Mixed ANOVA), the number of data sets included differs depending on the variable, because of excluded outliers.

The exclusion criteria of the study was the fulfillment of the WHO recommendations for PA for people aged ≥ 65 at baseline based on self-report and not by using objectively accelerometer data. Nevertheless, some participants evidently underestimated their own weekly amount of PA before participating in the study. When wearing the accelerometer, the amount of PA could be quantified. A subsequently exclusion of the very active participants (≥150 minutes moderate PA | ≥75 minutes vigorous PA) appeared to us to be inappropriate from a methodological point of view.

We defined all inclusion and exclusion criteria of the study a priori. The self-assessment was about the degree of time of weekly moderate physical activity was included to address a possible ceiling effect of PA in this age group. The operationalization of this exclusion criteria focused on self-reported degree of weekly moderate PA prior to recruitment, and therefore independent of the outcomes of the study. Thus, study participants whose self-reported PA exceeded the WHO recommendation of ≥ 150 minutes of moderate PA (objectively measured with an accelerometer device), were not excluded from the main analysis and only used for a sub-group analysis. 

Comment: The authors say: "To examine the effects of the intervention, we analyzed the overall, light, moderate and vigorous PA and PI after 6 months in the intervention group compared to the control group". These qualitative levels of PA have not previously been defined.

Response: In response to your concerns about the qualitative levels of PA, we moved the following sentences in the measures section and added explicit information to the manuscript: “We used specific cut points based on Freedson to categorize the intensity of participants’ PA. For adults, this cut-point model is considered as established for the determination of different physical activity categories. Intensity of PA is categorized as sedentary (0–99 counts), light (100–1951 counts), moderate (1952–5724 counts), and vigorous (5725–9498 counts) PA. Non-wear time was defined as ≥ 60 min of continual zero counts.” (please see lines 188 – 192)

Comment: The difference between intervention and non-intervention should be made explicit. Was it only that the experimental group received the letters and the control group did not receive them? According to the Declaration of Helsinki regarding placebo use, what was done with the control group at the end of the study?

Response: Thank you for this comment. The focus of our study was to examine the effect of a low threshold intervention. Only study participants in the intervention group received two individual feedback letters containing objectively measured PA and PI times based on data from the accelerometer device captured at baseline and 3 months follow-up measurement. Thus, this was the difference between the experimental (intervention) and control group. 

Participants in the control group did not receive any feedback while study time. According to the Declaration of Helsinki, we provided participants in the control group a detailed feedback about PA and PI after the end of the study within a thank you letter. This procedure ensured that all participants (both groups) received in sum the same information at the end of the study. In conclusion, there were no differences in type, access and scope of information between both groups (intervention and control). 

In response to your concerns about the explicit difference between the intervention and control group, we added the following sentence to the manuscript in the intervention section: “The participants in the control group received no feedback during the study period. According to the Declaration of Helsinki, we provided participants in the control group a detailed feedback about PA and PI after the end of the study within a thank you letter. This procedure ensured that all participants (both groups) received in sum the same information at the end of the study.” (please see lines 197 – 201)

Comment: Please specify what the variables "education" and "health status" measure.

Response: The education variable measured the number of years participants went to school (< 10 years, = 10 years, > 10 years). To address your concern, we specified the variable in the revised version of the manuscript and added the following sentence to the measures section: “number of years participants went to school”. (please see line 175)

In this manuscript, we did not include health status as a variable or analyzed it separately. For this reasons, we excluded the term “health status” from the manuscript. We now provide a more streamlined version of the measures section. Thank you for this comment. 

Comment: I agree with the reviewers that statistical analysis is not appropriate. Reviewer 2 suggests applying ANOVAs and considers it of interest to explore the effect of other variables (age, sex, etc.). Please report differences in age, sex, years of education, BMI, waist and hip circumference, blood pressure, etc., between groups; maybe it will be necessary to include some of these variables as covariables to mitigate their effects.

Response: Thank you for this comment to the statistical analysis. In the revised version of the manuscript, we provided a new statistical analysis: Two-Way Mixed ANOVA with repeated measures that is in line with reviewer 2. To address your concerns about the effect of other variables, we added the variables waist, hip circumference, blood pressure and pulse in the table 1 (descriptive characteristics of the study sample) that is consistent with Reviewer 2 recommendations (please see page 9/10). In addition, we checked all variables regarding significant group differences. Here, we detected a systematic difference in the variable “partnership” between intervention and control group. Therefore, we added the variable “partnership” as a covariate in the Two-Way Mixed ANOVA to mitigate the effect. For the ANOVA, we checked all assumptions and requirements (please see details below). 

In the revision, we added the new data file including the added variables (waist, hip circumference, blood pressure and pulse) and data sets as an attached file (please see S1_File.xls). 

Comment: I also agree with Reviewer 2 concerning some conclusions that are not supported by the results.

Response: In response to your concerns about the conclusion of the results, we adapted the conclusion section. Specifically, we now provide a more streamlined conclusion based on our results that is consistent with Reviewer 2 recommendations i.e. we deleted the following sentence “Low-threshold interventions have the potential to increase PA and reduce PI in the general elderly population.”

 

Reviewer #1: I will focus on methods and discussion

Major

1) As the authors acknowledge this is a select population that makes generalisability of the findings questionable.

Response: Thank you for your comment to the generalizability of the findings. Our analysis based on a convenience sample of participants. We used a variety of sampling methods, including the possibility of self-recruitment, some of which have likely increased the proportion of participants with above average PA compared to PA at the population level. We observed high levels of PA, particularly among the older age groups and females, which indicate some selection bias. For this reasons, the composition of our study sample somewhat limits the generalizability of our results. 

In our opinion, we already address your concerns about the generalizability of the findings in the limitation section: “Some of these methods have likely increased the proportion of participants with above average PA compared to PA in the general population – so the composition of our study sample somewhat limits the generalizability of our results.” (please see line 386) 

2) Some information on the randomisation procedure and how it was implemented is needed.

Response: Thank you for your comment about the random allocation. After screening and accelerometry, participants were individually randomized 1:1 in the intervention and the control group using an automated function (algorithm) in the documentation and study software. The design of the MOVING-study is published (Kleinke et al. 2018). In the trial, we provide information about the randomization process. In response to your concerns about the randomization, we added the following sentence to the manuscript: “The randomization was conducted using an automated algorithm in the documentation software. The documentation software based on electronic case report forms (Kleinke et al. 2018). 

3) The sample size has not been arrived to through a power calculation but it is a convenience sample.

Response: The design of the MOVING-study is published (Kleinke et al. 2018). In this paper, we provide detailed information about the power calculation as the following: “The sample size estimate was based on the amount of PA (mean time in min per day), measured in a comparable group of elderly people. In the study in question (Troiano et al.) the amount of PA was on average 15 min of combined PA (moderate and vigorous PA) per day. We took these data as the starting point for the estimation of the required sample size. We assume that participants in the control group remain at approximately the same level while people in the intervention group increase their amount of PA by 20%, which corresponds to an increase from 15 to 18 min of PA per day after 6 months. To demonstrate this effect, a total number of 151 participants is needed (standard deviation 6.0 min, alpha = 0.05, power = 0.80). Assuming a loss to follow-up of about 30%, we will recruit 216 participants.” (Kleinke et al. 2018)

The number of subjects we aimed in the study design (n = 151) was reached (n = 166) in order to prove an effect of the intervention. In the beginning of the methods section, (please see lines 144 - 145), we refer to the design paper of the study for more background information about the methods and study design. 

4) Results are poorly reported. Specifically the authors needs to make clear:

a) Drop outs - from 206 to 157?

Response: Thank you for this comment. For the traceability of the dropouts for each study group (with reasons), we provide a Consort diagram (Fig. 1) according to PLOS ONE requirements. The Consort diagram is an attached file (supporting information) – please see Fig. 1 “S1 Fig. Consort diagram: Recruited participants and participants included in the analysis (n = 166)”. Please see also the comment to the adapted sample size in the section for the editor. 

b) report characteristics by group (i.e. Table 1). was balance across these key parameters achieved?

Response: Thank you for your constructive comment to the statistical analyses. In response to your concerns about the descriptive statistics, we now provide a comparison between both groups (at baseline) that is consistent with Reviewer 2 recommendations. (please see table 1 “Descriptive characteristics of the study sample (n = 166), complete and valid data sets (baseline, 3 and 6 months follow-up”)

In addition, we checked all variables about significant differences and detected one significant difference in the variable “partnership” (p = 0.006). In the remaining variables, we detected no significant differences between both groups. Due to the significant difference in partnership, we included partnership as a covariable in the Two-Way Mixed ANOVA.

c) if not (i.e. b) then a t-test is not the apprpriate analysis

Response: Thank you for the comment. For baseline data, we analyzed the equality of variables. We considered your concerns about the statistical analyses and adapted our analyses. Specifically, we now performed a Two-Way Mixed ANOVA with repeated measures that is consistent with Reviewer 2 recommendations. In addition, we included partnership as a covariable in the analyses. First, we checked all assumptions and requirements (i.e. assumption of normality) for Mixed ANOVA. Please see a detailed description of the analysis of the assumptions below (Reviewer 2 section). Here, we provide statistical tests and figures for the analysis of the assumptions. 

e) a secondary outcome should have been time wearing the device and that should have been reported in the abstract as well

Response: Thank you for this comment. In response to your concerns about the wearing time, we added the following sentence to the abstract: “At baseline, the participants had a mean wearing time of 5,934.5 minutes (SD = 789.5) per week, which corresponds to about 14 hours daily on average.” (please see the result section in the abstract). In addition, we also report wearing time the device (at baseline, 3- and 6-months follow-up) in table 1 (study description). We defined all primary and secondary outcomes of the study a priori. A subsequent adjustment of the secondary outcomes appeared to us to be inappropriate from a methodological point of view.

f) report drop outs by group. Also the number of drop outs is relatively large for a keen sample of healthy adults.

Response: For the traceability of the dropouts for each study group, we provided a Consort diagram (Fig. 1) according to PLOS ONE requirements. The Consort diagram is an attached file (supporting information), in which the number of participants and the Lost to follow up is represent by Baseline, 3 and 6 months follow-up. Please see Fig. 1 “S1 Fig. Consort diagram: Recruited participants and participants included in the analysis (n = 166)”

At baseline, study participants were on average about 71 years old. Although, most participants showed a physically active lifestyle, people (and their partner) of this age group have often underlying diseases. The oldest participant was 85 years old. For this reason, in our opinion the dropout rate in the study is not surprisingly large. We published the study design (Kleinke et al. 2018), in which we assumed a dropout rate of about 30 percent in the power calculation. Due to this, the planned value of the dropouts reflects almost our experiences. In sum, we met the number of participants planned in the power calculation (n = 151) in our published study trial in 2018. 

5) No mention of missing data. was multiple imputation considered to boost the sample and get more robust estimates, compared to a complete case analysis?

Response: Thank you for your comment to missing data. After a carefully check of the data, we could increase the number of complete data sets about n = 5. Due to Intention-to-treat principle, we also included n = 4 incomplete datasets. In sum, we could increase the study sample about n = 9, so the study sample in the revised manuscript is now n = 166 at baseline. Due to the small number of incomplete data sets at Follow-up 1 (n = 1) and Follow-up 2 (n = 3), we decided us against an imputation of data. For the main analysis (ANOVA), the number of data sets differ depending on the depended variable, because of the exclusion of outliers. 

 

Reviewer #2: Authors evaluated the effectiveness of a low-threshold intervention to increase PA in elderly. Two groups (intervention-control) were recruited. No differences in most of the variables were found. Authors concluded that PA could be improved in participants with low levels of PA in both groups of this low-threshold intervention. Several concerns regarding the rationale, methodology, results and discussion are explained below.

Authors propose “targeting and tailoring are two important aspects of successful interventions” on basis of the participants’ needs and beliefs, however, what they propose here “a low-threshold intervention” does not meet this criterion. Please justify and show evidence (if the case) of the effectiveness of a more “practicable intervention”, maybe from studies were only biofeedback is used to increase PA.

Response: Thank you for the comment. You are concerns about the design of the intervention are right, our intervention design did not use targeting and tailoring due to the rationale. The rationale of the study was a low-threshold and practicable intervention aiming an increase of PA and a reduction of PI without using complex interventions including telephone/ video calls or personal consultations. In the introduction section, we refer to a systematic review with a positive effect using pedometers. To provide a more streamlined version of the manuscript, we moved this section to the discussion section.

Table 1. Why the number of participants (n) change across the descriptive characteristics, for instance, there are more (161) participants in the line “Number of participants currently living in a partnership (yes)” than those included in the study.

Response: Thank you for your response. The number of participants currently living in a partnership was wrong. Due to the increased number of the study sample (n = 166), we adapted all numbers of the descriptive statistics and removed the mistake. Thank you for that note.

We corrected the mistake and changed the number of participants currently living in a partnership to n = 160. In addition, the difference in the descriptive statistics can be justified by the fact that not every participant answered to all variables in the questionnaire at baseline (i.e. only 160 of 166 participants answered the question about the partnership). 

Please, include a Table showing the “descriptive characteristics” between groups and the statistical comparisons to exclude the presence of significant differences in these variables.

Response: Thank you for your constructive comment to the statistical comparison. In response to your concerns about the descriptive statistics, we now provide a comparison between both groups (at baseline) that is consistent with Reviewer 1 recommendations. (please see table 1 “Descriptive characteristics of the study sample (n = 166), complete and valid data sets (baseline, 3 and 6 months follow-up”). In addition, we checked all variables in table 1 on significant differences and detected a significant difference in the variable “partnership” (p = 0.006). Due to the significant difference in partnership, we included partnership as a covariable in the Two-Way Mixed ANOVA to mitigate the effect. 

Explain, in the exclusion criteria the cutoff for the fulfillment of the PA according to WHO that you considered. In this regard, if this was an exclusion criterion, why the whole sample was included in the first analysis, and just excluded for the subsequent subgroup analysis?

Response: 

Thank you for the comment. For the PA cutoff, we used the age-specific recommendation for people aged ≥65 years from the WHO, which is exactly the age of the study participants (age of ≥65 years). To address your comment, we added the following information to the manuscript about the specific cutoff for the fulfillment of the WHO recommendation: “(≥150 minutes of moderate PA or 75 minutes of vigorous PA).”

We defined all inclusion and exclusion criteria of the study a priori. The self-assessment was about the degree of time of weekly moderate physical activity was included to address a possible ceiling effect of PA in this age group. The operationalization of this exclusion criteria focused on self-reported degree of weekly moderate PA prior to recruitment. In contrast, the analysis focused on the amount of PA objectively measured with the accelerometer device. 

Explain, in data analysis, how delta values were calculated.

Response: We followed your recommendation to the statistical analysis and calculated in the revised version of the manuscript a Two-Way ANOVA with repeated measures. Due to the adjustment of the statistical analyses, we have not included delta in the analysis anymore. Due to this, we deleted the analysis of the delta to provide a more streamlined version of the manuscript. 

At methodological level, I see two main problems, 1) participants with ≥ 150 minutes of weekly moderate PA at baseline (according to the WHO recommendation) which actually fulfillment an exclusion criterion, were included. 2) The statistical approaching is not justified for this data, why to use t-test instead of two way anova with group (2) and follow-up (3) as factors? The latter could help you to understand within subjects and between groups differences. On the other hand, why data of the first follow-up is not analyzed but included in Tables and Figures?

Response: Thank you for the constructive comments to the statistical analyses that will help to increase the quality of the manuscript. 

1) The self-assessment was about the degree of time of weekly moderate physical activity was included to address a possible ceiling effect of PA in this age group. The operationalization of these exclusion criteria focused on self-reported degree of weekly moderate PA prior to recruitment.

We defined all exclusion criteria of the study a priori. We based the main analysis on the whole sample of the study participants irrespective of the results of the measurements within the study. The primary endpoint of our study however was the absolute number of minutes. For the main analysis we did not apply any threshold (i.e. 150 minutes of PA) to avoid any possible bias. However, we nevertheless evaluated the effect of the very active subgroup on the overall intervention effect in a sensitive analysis. In the revised version, we now explicitly report these outcomes. In addition, we discuss the inclusion of the very active subgroup in the limitations. Therefore, we added the following part to the limitations: “Prior to recruitment, we tried to exclude very active people from the study (based on self-assessment) to address a possible ceiling effect. As the results have shown, many participants underestimate their own amount of PA. Therefore, we calculated a sensitive analysis excluding very active participants with an amount of ≥150 minutes of moderate PA. Some ceiling effect reduced the overall intervention effect somewhat.” (please see limitation section)

2) We followed your recommendation to the statistical approach. In the revised version of the manuscript, we calculated a Two-Way Mixed ANOVA with time group (between subject) and time (inner subject) as factors. Due to a significant difference in partnership between intervention and control group at baseline (p = 0.006), we included partnership as a covariable in the Two-Way ANOVA to mitigate a possible effect. In the analysis we detected a significant interaction effect of time*partnership. For the ANOVA, we checked all assumptions and specific requirements. Please see a detail description of the check of the assumptions below. Here, we provide statistical tests for the verification of the assumptions (i.e. Shapiro-Wilk for verification of assumption of normality) and g-g-plots.

Due to adjustment of statistical analysis, we included the 3-months follow-up (first follow-up) in the ANOVA by using time as the independent variable. In addition to the inferential statistics, we analyzed all data using descriptive statistics. 

 Response: Thank you for the comment to the main analyses. In the revised version of the manuscript, we calculated a mixed ANOVA with time and group as factors and adjusted for partnership. For the ANOVA as a parametric test, crucial assumptions should be checked. Before the main analyses, we first checked the assumptions of the ANOVA beginning with the assumption of normality:

Assumption of normality 

First, we checked the assumption of normality for all dependent variables using Shapiro-Wilk Test, because the Shapiro-Wilk-Test has compared to the Kolmogorow-Smirnow-Test a higher statistical power (Razali & Wah, 2011; Steinskog, Tjøstheim & Kvamstø, 2007). In the analyses we detected in two dependent variables an equal distribution of the variables (light and overall PA), and in three dependent variables a non-normal distribution (PI, moderate & vigorous). Light and overall PA were normally distributed for both groups as assessed by the Shapiro-Wilk test (p > .05). Although, assumption of normality is violated in three dependent variables, ANOVA is a robust method against a violation of the assumption (Glass, Peckham, & Sanders, 1972; Harwell, Rubinstein, Hayes, & Olds, 1992; Salkind, 2010). In addition, our sample size is more than n = 30 in each study group. For this reasons, a transformation of the data is not necessary. As a result of the check of the assumptions, we added the following text to the manuscript (please see page 9): “We analyzed the assumptions and requirements of mixed ANOVA i.e. distribution and homoscedasticity and detected no violation of the assumptions”. 

Please see attached results of the Shapiro-Wilk-Test for light PA as an example for the analyses of normality. Light PA were normally distributed for both groups as assessed by the Shapiro-Wilk test (p > .05) (please see Tab. 1).

Tab. 1: Shapiro-Wilk-Test for Light PA

In addition to the Shapiro-Wilk-Test, we analyzed the distribution using Q-Q-Plots as you can see in Fig.1 for light PA in the intervention group and Fig. 2 in the control group.

Fig 1: Q-Q-Plot for light PA in the intervention group

Fig 2: Q-Q-Plot for light PA in the control group

In a second example, you can see the result of the Shapiro-Wilk-Test of a non-normal distributed variable. PI was not normally distributed for both groups as assessed by the Shapiro-Wilk test (p < .05) (please see Tab. 2).

Tab. 2: Shapiro-Wilk-Test for PI

Literature 

1. Glass, G. V., Peckham, P. D., & Sanders, J. R. (1972). Consequences of Failure to Meet Assumptions Underlying the Fixed Effects Analyses of Variance and Covariance. Review of Educational Research, 42(3), 237–288. doi:10.3102/00346543042003237

2. Harwell, M. R., Rubinstein, E. N., Hayes, W. S., & Olds, C. C. (1992). Summarizing Monte Carlo Results in Methodological Research: The One- and Two-Factor Fixed Effects ANOVA Cases. Journal of Educational and Behavioral Statistics, 17(4), 315–339. doi:10.3102/10769986017004315

3. Salkind, N. J. (2010). Encyclopedia of Research Design (Vol. 2). Los Angeles: Sage.

4. Razali, N. M., & Wah, Y. B. (2011). Power comparisons of Shapiro-Wilk, Kolmogorov-Smirnov, Lilliefors and Anderson-Darling tests. Journal of Statistical Modeling and Analytics, 2(1), 21-33.

5. Shapiro, S. S., & Wilk, M. B. (1965). An analysis of variance test for normality (complete samples). Biometrika, 52(3/4), 591-611.

6. Steinskog, D. J., Tjøstheim, D. B., & Kvamstø, N. G. (2007). A cautionary note on the use of the Kolmogorov-Smirnov test for normality. Monthly Weather Review, 135(3), 1151-1157.

Check for outliers in the data set

In a second step, we systematically detected outliers (for each dependent variable). Outliers were defined as any data point that is more than 1.5 times (light outliers) or more than 3 times of the interquartile range (IQR) (extreme outliers). We excluded outliers (data points > than 1.5 times of the IQR) from the analyses, because ANOVA as a parametric test is not robust against outliers. In addition, we wanted to evaluate the intervention effect in the general population. In our study sample, the amount of PA is already overestimated (i.e. one participant achieved about 800 minutes moderate PA per week). The number of excluded datasets is little. 

Due to the check of assumptions, we excluded different datasets regarding the dependent variable. Thus, the number of included data sets slightly changed between the Two-Way Mixed ANOVAS. 

After excluding outliers, the depending variables PI, light and overall PA were normally distributed. In addition, we also checked the homogeneity of the error variances (assessed by Levene’s test) and covariances (by Box’s test) and detected no violation of the assumptions. 

The only significant result was a difference in delta value between groups, however, this result did not confirm a PA increase due the intervention, but a decrease in the PA in the control group, according with Figure 2, is that right? Therefore, the effectiveness of the intervention is not corroborated?

Response: In the revised version of the manuscript, we changed the statistical method according to your recommendations. Therefore, we have not evaluated delta values any more and deleted it from the manuscript. 

In the first paragraph of the discussion, authors stated that the intervention group showed more overall PA compared to the participants in the control group, however this result was not significant. Therefore, no differences between groups were observed at any time.

Response: Thank you for the comment. Please see comment below.

In lines 286-287 authors suggest “Interventions should focus on the target group and especially on individual factors (e.g. age, sex)”, but not analysis was carried out with these variables in this study, it could be an interesting approaching. It is known, in previous literature, the differences between sexes in terms of PA and self-efficacy to PA.

Response: In a previous paper, we already analysed the determinants for PA and PI in a group of healthy elderly people separated by sex (Kleinke, 2020). Here, we found i.e. a statistical significant positive effect of self-efficacy in women (p = 0.020). Due to significant group difference, we included partnership as a covariate in the ANOVA. To address your concerns about the intervention design, we added the following text to the manuscript: “As previous research has shown”. (please see line 350 - 351) In this paper, we wanted to evaluate a low-threshold intervention to improve overall PA and reduce PI. 

From lines 301-313 authors discuss the importance of light PA increase and PI decrease, but their results did not show evidence of the effectiveness of the intervention even for these levels of PA.

Response: Although, there was no significant intervention effect on light PA, people aged over 65 years are a relevant target group especially for light PA. Independently and as the descriptive results show, study participants spent most of the time in light PA. Thus, interventions for the elderly should especially focus on light PA. 

In the conclusion authors suggest “Low-threshold interventions have the potential to increase PA and reduce PI in the general elderly population” however their results did not support this statement.

Response: Thank you for the comment. We deleted the sentence from the manuscript and tried to focus our statements which are supported by the results. In the revised version, we addressed your comment as the following: “Our analyses shows limited and not significant, but seems to have a small positive effect of a low-threshold intervention after 6 months follow-up.” (please see conclusion section). 

Literature (1–4)

1. Stamatakis E, Ekelund U, Ding D, Hamer M, Bauman AE, Lee I-M. Is the time right for quantitative public health guidelines on sitting? A narrative review of sedentary behaviour research paradigms and findings. Br J Sports Med [Internet]. 2018; Available from: https://bjsm.bmj.com/content/early/2018/06/08/bjsports-2018-099131

2. Kleinke F, Schwaneberg T, Weymar F, Penndorf P, Ulbricht S, Lehnert K, et al. MOVING: Motivation-Oriented interVention study for the elderly IN Greifswald: study protocol for a randomized controlled trial. Trials. 2018 Jan;19(1):57. 

3. Kleinke F, Penndorf P, Ulbricht S, Dörr M, Hoffmann W, van den Berg N. Levels of and determinants for physical activity and physical inactivity in a group of healthy elderly people in Germany: Baseline results of the MOVING-study. PLoS One [Internet]. 2020 Aug 13;15(8):e0237495. Available from: https://doi.org/10.1371/journal.pone.0237495

4. Ekelund U, Tarp J, Steene-Johannessen J, Hansen BH, Jefferis B, Fagerland MW, et al. Dose-response associations between accelerometry measured physical activity and sedentary time and all cause mortality: systematic review and harmonised meta-analysis. BMJ [Internet]. 2019;366. Available from: https://www.bmj.com/content/366/bmj.l4570

---

## [Decision Letter · Decision Letter 1]

27 May 2021

PONE-D-21-04633R1

A low-threshold intervention to increase physical activity and reduce physical inactivity in a group of healthy elderly people in Germany: Results of the randomized controlled MOVING study

PLOS ONE

Dear Dr. Kleinke,

Thank you for submitting your manuscript to PLOS ONE. After careful consideration, we feel that it has merit but does not fully meet PLOS ONE’s publication criteria as it currently stands. Therefore, we invite you to submit a revised version of the manuscript that addresses the points raised during the review process.

Very few corrections are missing for the acceptance of the article. Please provide the meaning of each line color in Figure 2, correct the use of parentheses and square brackets in the results section, and complete Reviewer 2's suggestions where possible.

Please submit your revised manuscript by June 7th 2021. I await your response very soon! Although if you will need more time than this to complete your revisions, please reply to this message or contact the journal office at plosone@plos.org. Please include the following items when submitting your revised manuscript:

We look forward to receiving your revised manuscript.

Kind regards,

Thalia Fernandez, Ph.D.

Academic Editor

PLOS ONE

Journal Requirements:

Additional Editor Comments (if provided):

Dear Dr. Kleinke,

Very few corrections are missing for the acceptance of the article. Provide the meaning of each line color in Figure 2, correct the use of parentheses and square brackets in the results section, and complete Reviewer 2's suggestions where possible.

I await your response very soon!

Kind regards,

Thalía

Reviewers' comments:

Reviewer's Responses to Questions

**Comments to the Author**

1. If the authors have adequately addressed your comments raised in a previous round of review and you feel that this manuscript is now acceptable for publication, you may indicate that here to bypass the “Comments to the Author” section, enter your conflict of interest statement in the “Confidential to Editor” section, and submit your "Accept" recommendation.

Reviewer #1: All comments have been addressed

Reviewer #3: All comments have been addressed

2. Is the manuscript technically sound, and do the data support the conclusions?

Reviewer #1: Yes

Reviewer #3: Yes

3. Has the statistical analysis been performed appropriately and rigorously? 

Reviewer #1: Yes

Reviewer #3: Yes

4. Have the authors made all data underlying the findings in their manuscript fully available?

Reviewer #1: Yes

Reviewer #3: Yes

5. Is the manuscript presented in an intelligible fashion and written in standard English?

Reviewer #1: Yes

Reviewer #3: Yes

6. Review Comments to the Author

Reviewer #1: I am happy with the authors' responses. Some minor comments and perhaps advice for future work:

1) try to use a regression model over ANOVA. it is more flexible, easier to interpret and allows for controlling for numerous covariates.

baseline characteristics appear similar across the two groups. report as mean (95% CI) rather than M (95% CI) in the respective table. You could include comparisons at baseline between the two groups, although I don't think anything is significantly different.

sensitivity rather than sensitive analyses

in the absence of a main effect we don't tend to examine for interactions

Reviewer #3: Minor details:

The authors should provide the meaning of each line color in figure 2, and correct the usage of parentheses and brackets in the results section.

7. PLOS authors have the option to publish the peer review history of their article (what does this mean?). If published, this will include your full peer review and any attached files.

Reviewer #1: No

Reviewer #3: **Yes: **Graciela Catalina Alatorre Cruz

---

## [Author Response · Author response to Decision Letter 1]

31 May 2021

PONE-D-21-04633

A low-threshold intervention to increase physical activity and reduce physical inactivity in a group of healthy elderly people in Germany: Results of the randomized controlled MOVING study

PLOS ONE

Response to Reviewers

Thank you for coordinating the review process and for your constructive comments to the manuscript. In our opinion, the comments help to increase the quality of the manuscript, especially the part of the statistical analyses. We have revised the manuscript again and addressed all reviewer’s comments from the minor revision. 

Reviewer #1

Comment: I am happy with the authors' responses. Some minor comments and perhaps advice for future work:

1) try to use a regression model over ANOVA. it is more flexible, easier to interpret and allows for controlling for numerous covariates.

Response: Thank you for this comment to the statistical analysis. In the revised version of the manuscript, we provide a Two-Way Mixed ANOVA with repeated measures that is in line with reviewer 2 from the major revision. For further analysis, we will take into account the advice from reviewer 1 and consider using a regression model. 

Comment: baseline characteristics appear similar across the two groups. report as mean (95% CI) rather than M (95% CI) in the respective table. You could include comparisons at baseline between the two groups, although I don't think anything is significantly different.

Response: At baseline, we checked all variables regarding significant group differences. Here, we detected one systematic difference in the variable “partnership” between intervention and control group. Therefore, we added the variable “partnership” as a covariate in the Two-Way Mixed ANOVA to mitigate the effect. All other variables in table 1 were not significantly different. In the revised version of the manuscript, we added the following sentence: “For baseline data, we analyzed the equality of variables between both study groups.” (please see lines: 210 – 211). 

We changed the report of table 2 that is consistent with Reviewer 1 recommendations (Mean (95% CI - please see green color in table 2).

Comment: sensitivity rather than sensitive analyses

Response: Thank you for this comment. In the revised version of the manuscript, we addressed the comment. 

Comment: in the absence of a main effect we don't tend to examine for interactions

Response: Thank you for this comment. We followed your recommendation and adapted the section with very few changes without reporting the details about the interaction effect. Now, we summarize the details about the interaction effect in one sentence: “P-values were ranging between 0.317 to 0.546.”. (Please see line: 260).

Reviewer #3

Comment: The authors should provide the meaning of each line color in figure 2, and correct the usage of parentheses and brackets in the results section.

Response: Thank you for this comment. The blue line represents the intervention group and the red line represents participants in the control group. We also added the meaning of each line in the manuscript (line 247). In the revised version of the manuscript, we have addressed Reviewer 3 comments and corrected the usage of parentheses and brackets in the results section (please see changes in green color and review mode in the manuscript).

---

## [Editor Report · Decision Letter 2]

24 Jun 2021

PONE-D-21-04633R2

A low-threshold intervention to increase physical activity and reduce physical inactivity in a group of healthy elderly people in Germany: Results of the randomized controlled MOVING study

PLOS ONE

Dear Dr. Kleinke,

Thank you for submitting your manuscript to PLOS ONE. After careful consideration, we feel that it has merit but does not fully meet PLOS ONE’s publication criteria as it currently stands. Therefore, we invite you to submit a revised version of the manuscript that addresses the points raised during the review process.

I consider that the article could be accepted after it satisfies the following requests:

1.- The authors should provide the meaning of each line color in figure 2.

2.- The authors should correct the usage of parentheses and brackets in the results section.

3.- Please report as mean (95% CI) rather than M (95% CI) in the respective table. 

4.- Please include comparisons at baseline between the two groups.

5.- Please, modify your wording: "sensitivity" rather than "sensitive analyses"

We look forward to receiving your revised manuscript.

Kind regards,

Thalia Fernandez, Ph.D.

Academic Editor

PLOS ONE
---

## [Author Response · Author response to Decision Letter 2]

6 Aug 2021

PONE-D-21-04633 

A low-threshold intervention to increase physical activity and reduce physical inactivity in a group of healthy elderly people in Germany: Results of the randomized controlled MOVING study

PLOS ONE

Response to Reviewers

Thank you for coordinating the review process and for your constructive comments to the manuscript. In our opinion, the comments help to increase the quality of the manuscript, especially the part of the statistical analyses. We have revised the manuscript again and addressed all reviewer’s comments from the minor revision. 

Reviewer #1

Comment: I consider that the article could be accepted after it satisfies the following requests:

1. The authors should provide the meaning of each line color in figure 2. 

Response: The blue line represents the intervention group and the red line represents participants in the control group. We also added the meaning of each line in the manuscript (please see lines 247, 249 and 608/609 in “manuscript_with_track_changes”). 

Comment: 2. The authors should correct the usage of parentheses and brackets in the results section.

Response: In the revised version of the manuscript, we have addressed Reviewer 3 comments and corrected the usage of parentheses and brackets in the results section (please see changes in green color and review mode in the manuscript). 

Comment: 3. Please report as mean (95% CI) rather than M (95% CI) in the respective table.

Response: In the revised version of the manuscript, we changed the report of Mean (95% CI) that is consistent with Reviewer 1 recommendations. Please see Table 2 in the manuscript. 

Comment: 4. Please include comparisons at baseline between the two groups.

Response: At baseline, we checked all variables regarding significant group differences. Here, we detected one systematic difference in the variable “partnership” between intervention and control group. Therefore, we added the variable “partnership” as a covariate in the Two-Way Mixed ANOVA to mitigate the effect. Other variables in table 1 were not significantly different. In the revised version of the manuscript, we added the following sentence: “For baseline data, we analyzed the equality of variables between both study groups.” (please see lines: 210 – 211).

Comment: 5. Please, modify your wording: "sensitivity" rather than "sensitive analyses"

Response: Thank you for this comment. In the revised version of the manuscript, we addressed the comment.

---

## [Editor Report · Decision Letter 3]

31 Aug 2021

A low-threshold intervention to increase physical activity and reduce physical inactivity in a group of healthy elderly people in Germany: Results of the randomized controlled MOVING study

PONE-D-21-04633R3

Dear Dr. Kleinke,

We’re pleased to inform you that your manuscript has been judged scientifically suitable for publication and will be formally accepted for publication once it meets all outstanding technical requirements.

Kind regards,

Thalia Fernandez, Ph.D.

Academic Editor

PLOS ONE
---

## [Editor Report · Acceptance letter]

3 Sep 2021

PONE-D-21-04633R3 

A low-threshold intervention to increase physical activity and reduce physical inactivity in a group of healthy elderly people in Germany: Results of the randomized controlled MOVING study

Dear Dr. Kleinke:

I'm pleased to inform you that your manuscript has been deemed suitable for publication in PLOS ONE. Congratulations! Your manuscript is now with our production department. 

Kind regards, 

on behalf of

Dr. Thalia Fernandez 

Academic Editor

PLOS ONE